# MicroRNA-Based Diagnosis and Treatment of Metastatic Human Osteosarcoma

**DOI:** 10.3390/cancers11040553

**Published:** 2019-04-18

**Authors:** Ryo Sasaki, Mitsuhiko Osaki, Futoshi Okada

**Affiliations:** 1Division of Pathological Biochemistry, Tottori University Faculty of Medicine, Yonago, Tottori 683-8503, Japan; sasary124@gmail.com (R.S.); fuokada@med.tottori-u.ac.jp (F.O.); 2Chromosome Engineering Research Center, Tottori University, 86 Nishi-cho, Yonago, Tottori 683-8503, Japan

**Keywords:** microRNA, osteosarcoma, lung metastasis

## Abstract

Osteosarcoma is a malignant tumor of the bones that commonly occurs in young individuals. The 5-year survival rate of osteosarcoma patients is 60–70%. Metastasis to the lungs leads to death in 30–40% of osteosarcoma patients. Therefore, the development of effective strategies for early detection and treatment of this disease are important to improve the survival of osteosarcoma patients. However, metastatic markers for osteosarcoma and molecules that might be targeted for the treatment of metastatic osteosarcoma have not been identified yet. Therefore, the mechanism of metastasis to the lungs needs to be explored from a novel viewpoint. Recently, the aberrant expression of microRNAs (miRNAs) has been reported to be involved in the carcinogenesis and cancer progression of many cancers. Furthermore, miRNAs in the blood have been reported to show an aberrant expression unique to several cancers. Therefore, miRNAs are gaining attention as potential diagnostic markers for cancers. On the other hand, normalizing the dysregulated expression of miRNAs in cancer cells has been shown to alter the phenotype of cancer cells, and thus treatment strategies targeting miRNAs are also being considered. This review summarizes the abnormality of miRNA expression associated with the metastasis of osteosarcoma and describes the present situation and issues regarding the early diagnosis and development of treatment strategies for metastatic osteosarcoma based on the current understanding of this disease.

## 1. Introduction

Osteosarcoma is a malignant tumor originating in the bones and is the most common among primary cancerous bone tumor arising in infants and adolescents, accounting for 10% of solid tumors developing in 15–19-year-old individuals [1,2,3]. Moreover, osteosarcoma occurring at 24 years of age and below is more common in males than in females at the rate of 3–5/million males compared to 2–4/million females [3]. The metaphyseal region of the long bones in the knee joints and vicinity of the shoulder joints are the most common sites for osteosarcoma [4]. In addition to surgery, the general treatments for osteosarcoma are chemotherapy and concomitant adjuvant therapy, and the overall 5-year survival rate of osteosarcoma patients is 60–70% [3,5]. Metastasis to the lungs is observed in 15–20% of osteosarcoma patients at the time of initial examination, and in such cases, the 5-year survival rate is 19%, which is extremely low [4,6]. Meanwhile, the 5-year survival rate of patients who are negative for metastasis to the lungs at the time of initial examination is 58–75% [7,8,9]. The prognosis of osteosarcoma patients is almost completely determined by metastasis as well as chemoresistance, particularly the status of metastasis to the lungs. Furthermore, micro-metastasis is estimated to be present in 60% of patients without obvious metastasis to the lungs at the time of initial examination [10]. Therefore, a highly sensitive diagnostic method for the early detection of micro-metastasis is essential to improve the survival of osteosarcoma patients.

Recently, microRNA (miRNA), which is present in body fluids, has been drawing attention as a new biomarker for the early diagnosis of various diseases. These miRNAs are a class of non-coding RNAs that are approximately 20 nucleotides in length. They are responsible for regulating post-transcriptional gene expression resulting in translational inhibition or degradation of the target mRNA [11,12]. At least 2,500 types of human miRNAs have been reported in miRBase (Release 22.1, www.mirbase.org), and the number is increasing annually. Since the initial discovery of miRNA in *Caenorhabditis elegans* (*C. elegans*) by Lee et al. in 1993, various physiological phenomena of miRNA and its involvement in human diseases have been reported [13]. In particular, many reports have stated the involvement of the expression of miRNA in the proliferation [14,15,16], migration [17,18,19], invasion [20,21,22], epithelial mesenchymal transition [23,24,25], and angiogenesis [26,27,28] of tumor cells, starting with the identification of miR-15 and miR-16 deletion in chronic leukemia by Calin et al. in 2002 [29]. In 2006, Volinia et al. showed that the miRNA profile of tumor tissues differed from that of normal ones in all analyzed malignancies; oncogenic miRNA (oncomiR) with an increased expression and tumor suppressor miRNA (tumor suppressor miR) with a decreased expression were reported to be present in tumor tissues [30].

It has been reported that miRNAs are present in a stable form in the blood [31]. Furthermore, Lawrie et al. reported that the expression of miR-21, -155, and -210 in serum derived from diffuse large B-cell lymphoma patients was increased compared to healthy volunteers. This was the first time that miRNAs in body fluids were shown to be useful as diagnostic markers for malignancies [32]. These miRNAs are secreted from cancer cells or cancer tissues by binding onto proteins or by being enveloped in extracellular vesicles to avoid degradation from RNase. Thus, if miRNAs expressed in higher quantities are secreted into the blood, then the detection of cancer can be determined by measuring the miRNA expression in the blood, meaning that the diagnosis of cancer from blood fluids can be expected.

As the aberrant expression of miRNAs results in carcinogenesis and cancer progression, they can be utilized as biomarkers for diagnosis and also can be targeted to develop cancer treatments. In other words, one can attempt to inhibit the carcinogenesis and progression of cancer by improving the aberrant miRNA expression either by inhibiting the function of miRNAs that are highly expressed or by supplementing miRNAs that are poorly expressed in cancers. In particular, cancer metastasis is a major barrier in cancer treatment, and thus it is important to develop new treatments or preventions against cancer metastasis based on the concept of miRNA dysfunction.

This review summarizes the aberrant expression of miRNA associated with osteosarcoma metastasis to the lung that has been reported to date, with a focus on human osteosarcoma. The possibility of utilizing miRNA for the diagnosis and treatment of metastasis as well as the current issues related to this prospect are also mentioned. 

## 2. Aberrant Expression of miRNA Related to Invasion and Metastasis of Osteosarcoma Cells

Osteosarcoma metastasizes to the lungs, bone, and lymph nodes. However, the main site is the lung, accounting for 80% of its metastasis [33,34]. Moreover, the 5-year survival of osteosarcoma patients with lung metastasis at the time of initial medical examination is as low as 19% [4,6], indicating that the status of metastasis to the lung is directly linked to the prognosis of osteosarcoma patients. For this reason, the mechanism of metastasis to the lung mediated by the aberrant expression of miRNA needs to be elucidated, and the early diagnosis and treatment based on that mechanism, established. Here, we introduce several molecules identified so far in osteosarcoma metastasis to the lung and the regulation of these molecules. 

Metastasis involves multiple steps as shown in Figure 1 [35,36]. Cancer cells in the primary lesion attract blood vessels and migrate into tissue to replenish their oxygen and nutrients. With time, they enter the blood vessels to invade the interior and subsequently adhere to the vascular endothelial cells of the target organ of metastasis via the blood circulation while avoiding attack from immune cells. They then infiltrate extravascularly from the blood vessel interior to form new tumor tissue in a distant organ or tissue; in other words, they form a metastatic lesion at the site. 

The angiogenesis accompanying tumor growth is for the purpose of obtaining oxygen and nutrient supply, and at the same time it increases the possibility of cancer cells infiltrating the blood vessel interior. Yuan-Ya et al. reported that in osteosarcoma cell lines (U2OS, MG63), decreased miR-374b expression increases the expression and secretion of vascular endothelial growth factor-A (VEGF-A), which acts on vascular endothelial cells, promoting angiogenesis [37]. In fact, the increased expression of VEGF-A in the primary lesion of osteosarcoma patients is correlated with an increase in the number of blood vessels, and this is reported to be involved in metastasis to the lung [38,39,40]. In addition, an inverse correlation was reported between miR-374b and VEGF-A expression in human osteosarcoma primary tumors [37]. Moreover, in osteosarcoma cell lines (MG63, 143B), the increased expression levels of angiopoietin 2, which is a regulator of angiogenesis as well as VEGF-A, have been shown to be triggered by a decrease in miR-543 expression. This has been reported to result in increased angiogenesis [41,42]. In primary lesion of human osteosarcoma, angiopoietin 2 expression is increased, and this has been reported to be correlated with progression of cancer via promoting distant metastasis [41]. Therefore, the aberrant expression of such miRNAs causes an increase in the expression levels of the angiogenesis factors (e.g., VEGF-A, angiopoietin 2), resulting in the angiogenesis of tumors.

Osteosarcoma cells migrate towards the induced neovascularization and infiltrate, while degrading the extracellular matrix (ECM), and with time they reach the blood vessels. There is type I collagen in bone matrix, and type IV collagen and laminin in the basement membrane of blood vessels as ECM proteins [43,44,45]. Matrix metalloproteinases (MMPs) are enzymes known to degrade the ECM, and 23 types have been reported in humans [46]. Of these enzymes, MMP-1, -2, -3, -8, -9, -13, and -14 have been reported to be involved in promoting the migration and infiltration of osteosarcoma cells [47,48,49,50,51,52,53,54]. The substrate of MMP-1, -8, and -13 is type I collagen, and that of MMP-2, -3, -9, and -13 is type IV collagen, and furthermore, laminin is another substrate of MMP-2, -3, and -14. Thus, increased expression of these MMPs promotes ECM degradation and is thus directly involved in infiltration. Jie et al. suppressed MMP-2, and -9 expression by introducing miR-218 into an osteosarcoma cell line (SaOS-2), resulting in the inhibition of migration and infiltration [55]. In addition, they showed that miR-218 expression decreased in primary osteosarcoma lesion [55]. Hui et al. reported that introducing miR-539 into an osteosarcoma cell line (MG63) inhibited expression of MMP-8 (which is a target gene of miR-539), resulting in suppression of the migration and invasion capabilities of the cell line [50]. Expression level of miR-539 also decreased in osteosarcoma primary lesion [50]. Osaki et al. reported that miR-143-3p expression decreased in a metastatic osteosarcoma cell line (143B) and primary osteosarcoma tissues with lung metastasis, resulting in an increase in the expression of MMP-13, which is a target gene of miR-143-3p, and hence, the invasion capability of the cell line increased [56]. Furthermore, Ezrin has been reported to be a molecule that is involved in the invasion of osteosarcoma cells [57,58]. Ezrin is part of the ERM (ezrin-radixin-moesin) family of proteins and is reported to be involved in the adherence of cancer cells, and their migration and invasion by cross-linking cellular membrane protein and actin filaments [59,60,61]. Junfeng et al. reported that the decreased expression of miR-183 correlated with the increased expression of Ezrin, which is the target gene in osteosarcoma cell lines (MG63, U2OS, SaOS-2, HOS) and in osteosarcoma primary lesion [62]. They showed that introduction of miR-183 into an osteosarcoma cell line (MG63) inhibited its migration and invasion [62]. Therefore, the increased expression levels of MMPs and Ezrin due to the aberrant expression of miRNA promote the migration and invasion of osteosarcoma cells and are thus involved in osteosarcoma progression.

Cancer cells that invade blood vessels circulate in the blood and disperse throughout the body. However, they need to avoid attack from host immune cells present in the blood, particularly cytotoxic T-lymphocytes (CTLs) that have anti-tumor effects. Because CTLs increase the expression of miR-23a by TGF-β stimulation, the expression of the target gene, BLIMP-1 (B lymphocyte-induced maturation protein-1) decreases, which is reported to cause a loss of the antitumor effect [63]. As osteosarcoma cells increase the expression and secretion of TGF-β [64], there is the possibility that the antitumor effect might be inhibited through increased expression of miR-23a in CTLs.

As the step in the final stage of metastasis, osteosarcoma cells circulating in the blood need to adhere to vascular endothelial cells of the target organ of metastasis. The CD146/MUC18 present on the surface of the cellular membrane of osteosarcoma cells is known as a molecule that is involved in intercellular adherence [65]. It is an important molecule that adheres osteosarcoma cells and vascular endothelial cells, and addition of anti-CD146 antibodies is reported to inhibit the adherence of both cell types [65]. To date, the miRNA that inhibits CD146 expression in osteosarcoma cells has not been identified. However, in vascular endothelial cells, miR-329 has been reported to target and inhibit the expression of CD146 [66]. Jiang et al. reported that miR-329 expression in human osteosarcoma tissue was decreased and that the level of expression was inversely related to the stage of cancer [67]. Moreover, the expression of CD146/MUC18 is higher in metastatic compared to non-metastatic patients in primary osteosarcoma tissue [68]. Thus, decreased miR-329 expression likely promotes the adherence of osteosarcoma cells to vascular endothelial cells by increasing CD146 expression.

## 3. Circulating miRNA in Blood as a Diagnostic Marker of Osteosarcoma Metastasis

In osteosarcoma, 15–20% of the patients are found to have lung metastasis at the time of initial medical examination. Meanwhile, micro-metastasis is estimated to be present in 60% of those patients without metastasis to the lung [10]. Currently, magnetic resonance imaging (MRI) and computed tomography (CT) combined with bone scintigraphy (BS) are used to determine the status of lung metastasis [69,70]. However, the number of metastases to the lung that can be detected by CT is two-thirds of the metastatic lesions that can be detected through surgical methods, indicating that not all the metastatic lesions can be identified [10]. In addition, small metastatic lesions that are ≤5 mm in size are impossible to detect by CT [10]. On the other hand, alkaline phosphatase (ALP) and lactate dehydrogenase (LDH) are being used as tumor markers of osteosarcoma in blood samples, but they cannot be detected in small tumors. In addition, these markers increase in not only osteosarcoma but also the other malignant tumors. They are not potent markers in terms of sensitivity and specificity. Therefore, there is a need for a highly sensitive method that is minimally invasive and which can be easily performed in order to detect micro-metastasis. Hence, liquid biopsy has been proposed as a new diagnostic method. Liquid biopsy is a method wherein diagnosis and treatment efficacy are measured qualitatively and quantitatively, by examining cells or molecules present in body fluids obtained from patients. Thus, the potential use of miRNAs as detection targets is gaining attention. The miRNAs that are secreted or leak from cancer cells are either present in vesicles formed by the lipid bilayer and are called extracellular vesicles, or form a complex with proteins, lipids, etc. and remain present in the blood without being degraded by RNase. This latter form of miRNA can be obtained by blood sampling, and the process is thus minimally invasive, and moreover, the miRNA can be collected repeatedly over time. In addition, the collected miRNA is not affected by freezing and thawing and, thus, consistent data can be obtained. In fact, Chen et al. confirmed that when miRNA derived from the serum of lung and colon cancer patients was frozen and thawed repeatedly for a total of 10 times, no degradation was observed [71]. In addition, Mitchell et al. reported that even when miRNA extracted from human serum was exposed to room temperature for 24 h, there was no degradation [72]. Thus, if plasma miRNA specific to osteosarcoma metastasis is identified, it will be useful and applicable for early diagnosis and periodical clinical examination. 

So far, it has not been identified plasma or serum miRNAs which could be a specific marker for osteosarcoma metastasis. If miRNA that is specific to osteosarcoma can be detected from serum and plasma derived from patients, it can be utilized in early diagnosis aimed at detecting metastasis. The miRNAs with aberrant expression that are present in serum or plasma derived from osteosarcoma patients, which have been reported to date, are summarized in Table 1. Compared to patients without metastasis, the miRNAs showing an increased expression in the plasma and serum of metastatic osteosarcoma patients were miR-27a, -195-5p, -199a-3p, and -221 [73,74,75,76,77]. Meanwhile, those showing a decreased expression were miR-34b, -95-3p, -133b, -195-5p, -206, -223-5p, -326, -375, and -491 [78,79,80,81,82,83,84,85]. Of these, miR-27a with increased expression and miR-95-3p and -195 with decreased expression were significantly present in the patients with lung metastasis, and the level of respective expression was not correlated with the size of the primary lesion [73,79,80]. Furthermore, the plasma levels of ALP and LDH, which have been used as conventional osteosarcoma biomarkers, showed no significant difference between non-metastatic patients and metastatic patients. Thus, miRNAs have been strongly demonstrated to be very useful markers in differentiating the status of lung metastasis. Data on these three miRNAs have been respectively reported as individual study results, and a combination of the expression levels of the three miRNAs could possibly be useful as metastatic markers with an even higher precision. In fact, Ochiya et al. combined the expression levels of multiple miRNAs as diagnostic markers of several cancers, including colon cancer and breast cancer, in order to construct a highly precise diagnostic method based on liquid biopsy [86,87,88]. Thus, it is necessary to examine the specific combination of multiple miRNAs to detect osteosarcoma metastasis. miRNAs other than miR-27a, -95-3p, and -195, have the potential to be used as osteosarcoma metastasis markers based on their increased or decreased expression levels. However, it is necessary to compare the altered expression of such miRNAs with existing markers and with clinical data in order to elucidate the significance of them. In addition, the methods for performing dosage compensation by quantitative analysis of miRNA are not necessarily uniform, and further investigations are needed. In the quest to develop early diagnostic methods that target plasma or serum miRNA, it is necessary to identify target miRNAs and their combination, specific to osteosarcoma metastasis, in order to develop detection methods with a high degree of specificity and sensitivity, taking into consideration normalization methods. Although U6 has been widely used as normalizer in serum miRNAs (Table 1), U6 is inappropriate in the normalization of serum miRNA levels because the expression of U6 is different in individuals and by age [89,90].

To achieve early detection and potentially inform early treatment of micrometastasis of osteosarcoma, higher grade detection with increased sensitivity will be required. However, it would be faced with two problems: (i) how to detect localization of the micrometastasis, and (ii) how to treat patients with the micrometastasis. Therefore, it is important to develop identification system(s) and treatment strategies for micrometastasis of osteosarcoma as well as a detection system using circulating miRNA signatures.

## 4. Developing Treatment Strategies for Metastatic Osteosarcoma Using Oligonucleotide Drugs

As mentioned above, exploration of novel biomarkers based on the aberrant expression of miRNA in cancer metastasis is progressing. Meanwhile, studies on the possibility of developing miRNA drugs that target the dysregulated expression are also advancing. In brief, there are two therapeutic approaches. The first is to inhibit the function of miRNAs with increased expression in cancer, and the second is to administer miRNAs showing decreased expression in cancer.

For the first approach, a method has been developed of treating complementary nucleic acids which can be bound to overexpressed miRNA to weaken its function. For example, (i) antisense miRNA oligonucleotides (AMOs) [91], (ii) miRNA sponge [92], (iii) miRNA eraser [93], and (iv) miRNA inhibitors using Tough Decoy constructs [94] have been reported. Antisense miRNA oligonucleotides are the most commonly used inhibitors of miRNA. Antagomir and locked nucleic acid (LNA) are also being used to inhibit miRNA expression. Antagomir is an artificial nucleic acid where the phosphate backbone is modified to a phosphorothioate, and the O in the 2′ position modified to OMe [95]. On the other hand, LNA is an artificial nucleic acid consisting of a conformation where the 2′ and 4′ positions of the pentose sugar of the ribonucleic acid are cross-linked with an -O-CH2- group, locking it into an N-type. It has a high ability to hybridize with target nucleic acid molecules [96]. Both LNA and antagomirs are resistant towards nuclease and are therefore stable. Furthermore, they are being modified to lower their cellular toxicity or potentiate the efficiency of their cellular uptake. Inhibitors other than AMOs, such as miRNA sponge, miRNA eraser, and those using Tough Decoy constructs, all suppress the function by triggering the absorption of specific miRNAs. Fujiwara et al. reported treatment efficacy using LNA for metastatic osteosarcoma [97]. Human osteosarcoma cell lines showing high CD133 expression have cancer stem cell-like properties and were found to have high miR-133a expression compared to cell lines with low CD133 expression. Furthermore, it was evident that the introduction of miR-133a to cells with low CD133 expression increased their infiltration capability. In contrast, inhibition of miR-133a using LNA decreased the infiltration capability of cells with high CD133 expression. Intravenous administration of LNA, which impaired miR-133a function, to an osteosarcoma model of spontaneous lung metastasis showed a significant decrease in metastasis to the lung. Therefore, miR-133a dysfunction is likely to inhibit osteosarcoma metastasis to the lung. Moreover, Jin et al. showed miR-135b highly expressed in osteosarcoma tissues. High miR-135b expression activated Wnt/β-catenin and Notch signaling pathways, which contributed to promoting tumor invasion and stemness in vitro, and lung metastasis in vivo. Dysfunction of miR-135b suppressed lung metastasis by using LNA [98].

The second approach is supplementing miRNAs downregulated in cancer by administrating synthetic miRNA mimics. However, it is difficult to modify or change miRNA structure without loss of the function, in contrast to AMOs that are based on inhibition of function. Therefore, its administration without any protection has limitations. Carrier molecules to ensure stability of these miRNAs are essential. To date, the development of methods based on peptides, such as atelocollagen, liposomes that have undergone several modifications and nanoparticles, such as exosomes, are progressing. We investigated the efficacy of a method of administering synthetic miRNA in preventing osteosarcoma metastasis to the lung [56]. We compared the miRNA expression profiles of a highly metastatic human osteosarcoma cell line (143B) with that of a non-metastatic cell line (HOS) and found that miR-143 is poorly expressed and down-regulates the invasion capability. The transfection of miR-143 to 143B cells in vitro inhibited the invasion capability without affecting cell proliferation. Subsequently, intravenous administration of miR-143 using atelocollagen as carrier to a model of spontaneous osteosarcoma lung metastasis resulted in inhibition of metastasis to the lung with no effect on proliferation of the primary lesion. In addition, miR-143 inhibits the invasion capability targeting MMP-13, and furthermore, it has also been confirmed that even in human clinical samples, MMP-13 expression is poor in cases where miR-143 expression is high. Furthermore, Xin et al. found that de novo lipid synthesis enhanced through the increase of ATP citrate lyase (ACLY) expression in osteosarcoma tissue. The miR-22 directly regulates ACLY expression. Further, there is the inverse association between miR-22 and ACLY expression in clinical samples of osteosarcoma. In vitro, transfection of miR-22 to an osteosarcoma cell line (Saos-2) suppressed growth and invasion, and promoted apoptosis. Moreover, injection of miR-22 using in vivo-jetPEI^®^ as carrier inhibited tumor growth and metastasis in animal model [99].

Along with the development of oligonucleotide drugs that target miRNA, it is also necessary to improve efficacy, specificity, and safety. Therefore, developing a drug delivery system (DDS) that is stable and appropriate for reaching the target site is important either when miRNA is dysregulated or when it has to be administered. However, for developing such a DDS, there are four issues that need to be addressed (Figure 2). The first is protection from RNase present in body fluids containing blood, as mentioned earlier. Thus, in addition to the modification of the RNA molecule per se, the development of carrier molecules or particles that will bind or embed nucleic acids is essential. The second is the fact that administered miRNA readily accumulates in the lungs, liver, and bone marrow and is filtered by the glomerulus of the kidney and excreted [100,101]. Therefore, a mechanism by which miRNA is delivered specifically to tumor cells is necessary. The third is the fact that the administered miRNA could be taken up by non-target cells prior to being delivered to the target cells. Hence, the desired effect could not be obtained and, instead, an off-target effect was displayed. In particular, phagocytosis by immune cells is regarded as a drawback, and measures to solve this have been reported. Kamerkar et al. reported that they produced a CD47-overexpressing human fibroblastic cell line (BJ), and collected extracellular vesicles derived from BJ cells with high CD47 expression. Binding of CD47 present on extracellular vesicles with SIRPα expressed on immune cells inhibited the phagocytosis by immune cells. This resulted in inhibition of the uptake of these extracellular vesicles by immune cells as well as increased delivery to cancer cells [102]. The fourth issue is the possibility of unnecessary spontaneous immune responses being induced. There are dendritic cells and macrophages in the blood, and as mentioned earlier, when miRNAs are taken up by these cells, toll-like receptors (TLRs)-3, -7, and -8 recognize single-stranded or double-stranded RNAs and induce inflammation. When double-stranded RNAs are taken up by macrophages, TLR-3 is activated, resulting in NF-kB activation and increased secretion of TNF-α and IL-6, reported to be involved in the proliferation of cancer cells [103], which could be a reverse effect. Fabbri et al. reported that the uptake of miRNA by macrophages activates TLR-7 and -8, resulting in NF-kB activation as well as increased secretion of TNF-α and IL-6, which caused an increase in cancer cell metastasis [104]. This likely induces cancer malignancy. 

In the treatment of human osteosarcoma lung metastasis, when miRNA is used as an oligonucleotide drug, identification of the target molecule as well as determining the efficacy of the drug need to progress collectively to achieve a highly specific delivery system.

## 5. Conclusions

This review summarized the association of the aberrant expression of miRNAs with osteosarcoma lung metastasis and described the possibilities and problems related to the diagnosis and treatment of this disease. All malignant cells, including osteosarcoma cells with acquired metastatic capability, have the ability to overcome multiple steps, as mentioned earlier. However, one can assume that the dysregulated expression of multiple miRNAs occurs at the same time. There is a need to identify an miRNA group that is specific to metastatic osteosarcoma, is secreted in the blood, and shows the aberrant expression that could be used as an early diagnostic marker. It is also required to identify an miRNA group showing aberrant expression that is broadly involved in the process of metastasis, which could be used in the treatment and prevention of osteosarcoma metastasis. Furthermore, the development of highly precise methods for the detection and delivery of miRNA molecules is also important, and advancements in diagnostic methods and drug development are expected in the future.

## Figures and Tables

**Figure 1 cancers-11-00553-f001:**
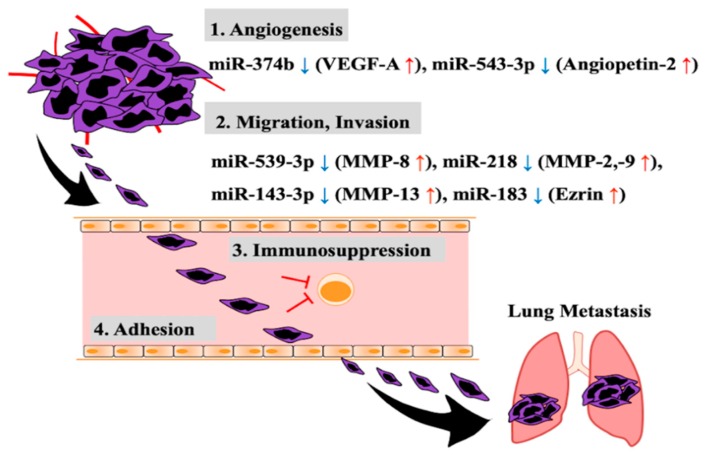
Dysregulated miRNAs and their target genes related to metastasis of osteosarcoma.

**Figure 2 cancers-11-00553-f002:**
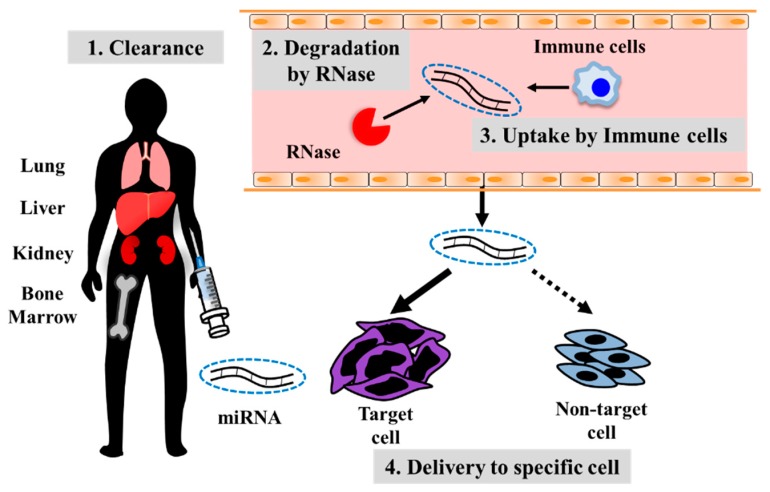
Challenges for miRNA-based cancer therapy development.

**Table 1 cancers-11-00553-t001:** miRNAs in serum or plasma derived from osteosarcoma patient.

miRNA	Expression	Normalization	Technique	Reference
miR-27a	Up	U6	qRT-PCR	[73]
miR-195-5p	Up	serum volume	qRT-PCR	[74]
miR-199a-3p	Up	serum volume	qRT-PCR	[74,75]
miR-221	Up	U6	qRT-PCR	[76,77]
miR-34b	Down	miR-39	qRT-PCR	[78]
miR-95-3p	Down	U6	qRT-PCR	[79]
miR-133b	Down	U6	qRT-PCR	[80]
miR-195-5p	Down	U6	qRT-PCR	[81]
miR-206	Down	U6	qRT-PCR	[80]
miR-223-5p	Down	U6	qRT-PCR	[82]
miR-326	Down	RNU48	qRT-PCR	[83]
miR-375	Down	U6	qRT-PCR	[84]
miR-491	Down	U6	qRT-PCR	[85]

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
