# Peer review of "MicroRNA-Based Diagnosis and Treatment of Metastatic Human Osteosarcoma"

_cancers, 2019, doi:10.3390/cancers11040553_

Round 1

Reviewer 1 Report

In their manuscript entitled „MicroRNA-based diagnosis and treatment of metastatic human osteosarcoma”, Sasaki and colleagues review the recent research and microRNAs and target genes involved in the metastasis of osteosarcoma and debate the applicability of metastasis-related microRNAs for osteosarcoma diagnosis via liquid biopsy or treatment via miR replacement or suppression. The review is informative and covers the recent literature, the conclusions are adequate. This is the first Revision of the manuscript.

The authors answered to all my critic Points in an adequate way. I do not have any further critics.

Best regards.

Author Response

I deeply appreciate your detailed review and comments regarding our paper. Your comments are very useful for our paper. 

Reviewer 2 Report

The authors adequately addressed my initial concerns. Please have the manuscript proofread for grammatical mistakes.

Author Response

I deeply appreciate your detailed review and comments regarding our paper. Your comments are very useful for our paper.

This manuscript is a resubmission of an earlier submission. The following is a list of the peer review reports and author responses from that submission.

Round 1

Reviewer 1 Report

In their manuscript entitled „MicroRNA-based diagnosis and treatment of metastatic human osteosarcoma”, Sasaki and colleagues review the recent research and microRNAs and target genes involved in the metastasis of osteosarcoma and debate the applicability of metastasis-related microRNAs for osteosarcoma diagnosis via liquid biopsy or treatment via miR replacement or suppression. The manuscript is informative and covers the recent literature, the conclusions are adequate. I have only one minor comment:

Although the manuscript is written understandable, there are still some passages, there proofreading would be advisable. For instant: l.20: an aberrant expression; l.21 improving à better use: normalizing (or another term); l. 24: abnormality à what should this mean?; l. 31-33: three time repeatedly used “occurring” à sharpen the sentences; l.48: which are;  l. 61: were reported; l.71: miRs expressed in higher levels; l.70-73: what should this mean? I do not fully understand this sentences; l.92: introduce several molecules identified so far; l. 212: What refers the particle “They” to? I think, it is to microRNAs, however, the sentence before the authors talk about LDH and ALP. Please specify.

Reviewer 2 Report

The manuscript addresses the use of circulating miRNAs as diagnostic biomarker, the involvement of miRNAs in cellular processes linked to metastasis in osteosarcoma cells and a rather speculative chapter on the therapeutic use of miRNAs. I am afraid the current manuscript lacks novel insights and clear future directions that make it a worthwhile read for interested cancer researchers and clinicians. Major comments: 1. Circulating microRNAs can be instrumental for the early detection of osteosarcoma, however, it is not clear whether metastasis – let alone micrometastasis - can be detected. Particularly the detection of micrometastasis may be of clinical value but is based on the assumption that the miRNA expression profiles of metastatic osteosarcoma are defined, and differ from the primary tumors. As pointed out in the manuscript there are miRNAs that affect metastatic processes in vitro using cell lines. Are there also data acquired from clinical tumor samples that clearly indicate that primary tumors and metastasis can be distinguished on the basis of their miRNA expression profiles? Are, for example, the miRNAs mentioned to be involved in migration and invasion differentially expressed in primary and metastatic tumors? 2. If one assumes micrometastasis can be detected through circulating miRNA signatures one still has to tackle the problem how to treat patients with micrometastasis. They are relative invisible to the standard imaging techniques excluding surgery and radiotherapy leaving chemotherapy as the sole treatment option. The authors should discuss these options in the manuscript. 3. A pitfall in using miRNAs as biomarker or for treatment purposes is tumor heterogeneity, what is known about heterogeneity in osteosarcomas? Are all metastasis the same? 4. The miRNAs listed in Table 1 are derived from different studies. What exactly is the overlap between the different studies, are the same miRNA detected in multiple studies? These miRNAs may make more reliable markers. Minor comments: 1. Figure 1 and Table 1 – please indicate for the mentioned miRNAs whether the 5p or 3p arm is involved? 2. Table 1 – miR-375 ‘Donw’ Please correct typographical error. 3. Page 5, lines 199-200 – Unclear what is meant by this sentence, please rephrase. 4. Page 6, line 233 – ‘administrate’ should be ‘administer’, please correct. 5. Page 7, line 258-259 – ‘However, it is difficult ….based on inhibition of function.’ What exactly is meant by this sentence, please rephrase.

Reviewer 3 Report

it is well written and documented revieww

Reviewer 4 Report

The manuscript “Micro-RNA based diagnosis and treatment of metastatic osteosarcoma” by Sasaki et al. provides a review of existing literature on the role of microRNAs in osteosarcoma metastasis, their potential use as serum biomarkers, and finally their potential therapeutic uses. In my view this review lacks sufficient depth to be considered for publication at this time. Many important studies have not been included, especially those that address microRNA expression in osteosarcoma tissue specimens.

There is potential to expand this review to more thoroughly, compare, contrast, and critique existing studies. In most cases studies are simply listed along with their results, the authors do not comment on the quality of the studies.  I believe a critical assessment of the various studies is very important as there are often conflicting in the microRNA literature due to major differences in the quality of studies evaluating microRNAs in cancer cells and tissues. I would also like to see more information from studies analyzing tumor tissue biopsies, the majority of the studies cited primarily examine conventional cell lines.

A large portion of this article reviews osteosarcoma epidemiology and pathogenesis. This information is likely to be familiar to most readers with an interest in this topic. I feel it would potentially strengthen this article to reduce the overview of osteosarcoma and to focus more on discussing existing miRNA literature relevant to osteosaromca.

Section specific comments and recommendations are included below:

Introduction

Line 35: “The metaphyseal region of the long bones in the knee joints 35 and vicinity of the shoulder joints are the most common sites for osteosarcoma, and the femur, tibia 36 and humerus are the three most frequent sites.” This sentence reads awkwardly to me, it seems strange to state the most common sites followed by the most frequent sites. I am not sure what the distinction here would be?

Line 42: “The prognosis of osteosarcoma patients is almost completely determined by metastasis, particularly the status of metastasis to the lungs.” Although metastasis is very important, response to chemotherapy is also a major predictor that I would mention here. It may be an over statement to say that prognosis is almost completely determined by metastasis.

Line 50: “They are responsible for regulating post-transcriptional gene expression resulting in translational inhibition or degradation of the target mRNA.” I would recommend changing the end of this sentence to say “target mRNAs” instead of target mRNA, as almost all microRNAs target more than one mRNA.

Line 63: “Although the aberrant expression of miRNA in tumor cells itself and tumor tissue has been reported, it seemed that plasma miRNA cannot be detected due to its biodegradability.” I would recommend removing the portion that states “it seemed that plasma miRNA cannot be detected due to its biodegradability” I am not aware of this ever being a concern for miRNA, it has long been established that miRNA are very stable.

Line 65: “Furthermore, Lawrie et al. reported that the expression of miR-21, -155, and -210 in serum derived from diffuse large B-cell lymphoma patients was increased compared to healthy volunteers. This was the first time to show that miRNAs were useful as diagnostic markers for malignancies.” I would consider revising this sentence. Although Lawrie et al may be one of the first published studies to show the value of miRNA in malignancy, I am not sure it was truly the first time that it was shown that miRNAs were useful diagnostic markers for malignancy. The concept had been under study much earlier than this.

Line 70: “Thus, if miRNAs expressed increased quantities are secreted into the blood, then, the detection of cancer can be determined by measuring the miRNA expression in the blood, meaning that the diagnosis of cancer from blood fluids can be expected.” This sentence contains grammatical errors, please consider revising.

Line 74:  “Since the aberrant expression of miRNAs results in the carcinogenesis and cancer progression, they can be utilized as biomarkers for diagnosis and also can be targeted to develop cancer treatments.” I would recommend removing the word “the” in front of carcinogenesis.

2. Aberrant expression of miRNA related to invasion and metastasis of osteosarcoma cells

Throughout this section the authors’ link miRNAs with gene targets associated with metastasis. In my view a single gene association is not sufficient to link a miRNA to having a regulatory effect on metastasis, as most microRNAs have many gene targets.  For example, the authors link miR-374b expression to VEGF-A expression, and miR-543 to angiopoietin-2 expression and thus conclude that these two microRNAs are involved in angiogenesis and osteosarcoma metastasis. microRNAs have many targets and many microRNAs target VEGF-A and angiopoietin-2 without impacting angiogenesis and metastasis. In my view the link between miR-374b and miR-543 is very speculative. In my view more evidence should be provided to show they are truly associated with metastasis or they should be removed from the review.

The discussion of miR-329 and its link to adhesion is interesting as there is a link to its targets and data studies showing it has decreased expression in osteosarcoma tissues.

Tumor size and anatomic location are also very important predictors of osteosarcoma metastasis. Large central osteosarcomas such as those involving the pelvis are much more likely to metastasize in comparison with those involving the distal extremities. It may be useful to comment on these issues and address how other studies have taken this into account in their analysis.

3. Circulating miRNA in blood as a diagnostic marker of osteosarcoma metastasis

I am not sure it is necessary to include so much information on CTCs and alternative non-miRNA biomarkers for osteosarcoma. This is an extensive topic, given the scope of the paper I would recommend that the author’s focus primarily on miRNA serum markers in this section. As currently written almost half of this section discusses non-miRNA biomarkers.

In regards to the previously published studies on serum miRNAs I would recommend that the authors discuss issues regarding normalization in greater detail. Many of the studies have used U6 as a normalizer, this can be a major problem in serum miRNA analysis as U6 is a large molecule and can degrade much faster than much smaller miRNA.

I would also recommend discussing other tissue specific miRNAs and their impact. For example miR-206 and miR-133 frequently show up as serum miRNAs associated with osteosarcoma. These are known muscle microRNAs, could osteosarcoma invasion into tissues such as muscle be causing the changes in serum miRNA expression?

4. Developing treatment strategies for metastatic osteosarcoma using oligonucleotide drugs

This section discusses theoretical miRNA therapeutics and cites possible efficacy based on miRNA targeting therapeutics in osteosarcoma cell lines. The described work is theoretical and very early stage. It is not clear if miRNA therapeutics will yet be viable for use in vivo. It would be useful to discuss the short comings of existing methods and possible steps to overcome these challenges.

This section also spends a great deal of space discussing possible techniques for therapeutically targeting miRNAs. Given the title of the article I would expect to hear more regarding what has been specifically published regarding the use of miRNAs to treat osteosarcomas rather than hearing about the various theoretical mechanisms for miRNA therapeutics.